# Tag Map: A Text-Based Map for Spatial Reasoning and Navigation with Large Language Models

**Mike Zhang, Kaixian Qu, Vaishakh Patil, Cesar Cadena, and Marco Hutter**
Robotic Systems Lab, ETH Zurich, Switzerland
`tag-mapping.github.io`

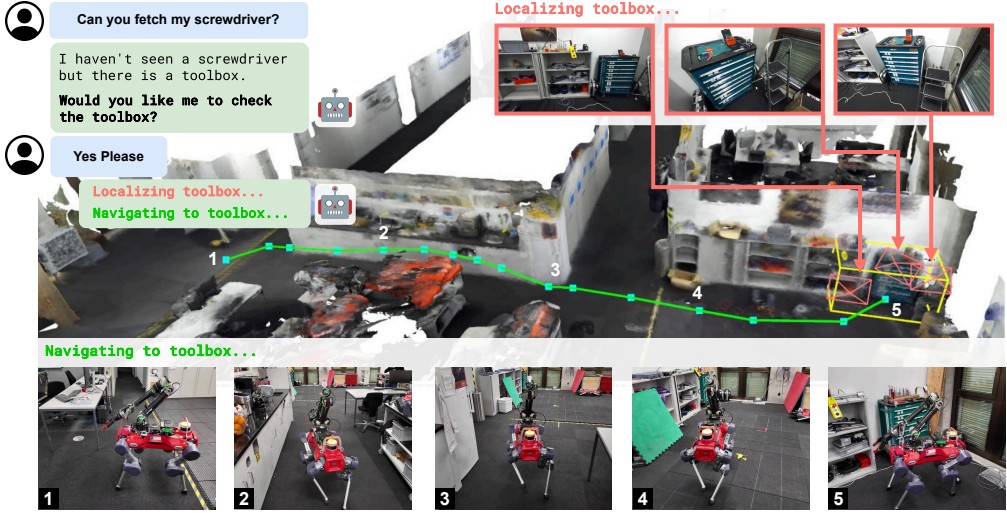

Figure 1: Our text-based map grounds an LLM to plan actionable navigation goals given the scene context to address user-specified tasks.

**Abstract:** Large Language Models (LLM) have emerged as a tool for robots to generate task plans using common sense reasoning. For the LLM to generate actionable plans, scene context must be provided, often through a map. Recent works have shifted from explicit maps with fixed semantic classes to implicit open vocabulary maps based on queryable embeddings capable of representing any semantic class. However, embeddings cannot directly report the scene context as they are implicit, requiring further processing for LLM integration. To address this, we propose an explicit text-based map that can represent thousands of semantic classes while easily integrating with LLMs due to their text-based nature by building upon large-scale image recognition models. We study how entities in our map can be localized and show through evaluations that our text-based map localizations perform comparably to those from open vocabulary maps while using two to four orders of magnitude less memory. Real-robot experiments demonstrate the grounding of an LLM with the text-based map to solve user tasks.

**Keywords:** Scene Understanding, Grounded Navigation, Large Language Models

## 1 Introduction

Scene understanding is fundamental for robots to generate actionable plans within their environment. A popular task for evaluating scene understanding is Object Goal Navigation (ObjectNav) [1, 2], where the robot must navigate to a target object within an unknown scene. Two components have emerged as recurring aspects in recent works on ObjectNav. First, a Large Language Model (LLM) which plans navigation goals, and second, a map that serves as a memory module storing observed objects and which can ground the reasoning of the LLM.

8th Conference on Robot Learning (CoRL 2024), Munich, Germany.

Map representations in previous works can be classified as either explicit or implicit. Explicit maps label locations as a class from a fixed set of classes [3, 4, 5]. This limits the tasks the map can handle but has the advantage of directly reporting the classes it contains. In contrast, implicit, or open vocabulary maps build upon the line of work in computer vision starting from CLIP [6], later extending to object detection [7, 8] and segmentation [9, 10, 11, 12], based on vision embeddings that are queryable by arbitrary text. Recent works in ObjectNav have considered maps where a location stores an embedding instead of a label [13, 14]. In theory, such maps can represent any objects, rooms, and affordances. In practice, there is no guarantee that the embeddings will capture the desired information. Moreover, embeddings cannot directly report what entities they represent. Instead, they must be queried to retrieve potential matches. For simple tasks along the lines of `"go to <thing>"`, such maps can be directly queried for `<thing>`. However, additional processing is needed to identify relevant queries for more complex tasks lacking an explicit goal object or region.

We propose a text-based map that can represent an extensive amount of semantic classes *explicitly* by leveraging multi-label image classification models trained to recognize thousands of semantic classes. Our map called the *tag map*, is a memory-efficient unstructured database storing viewpoints and the recognized entities for each viewpoint as text tags. Despite the minimal amount of information contained within, we find that tag maps can produce 3D localizations for both objects and rooms/regions with sufficient precision and recall to serve as useful navigation goals. Moreover, the explicit text nature of the tag map lends itself well for use in grounding LLMs.

Our main contributions are the following:

- A text-based map, building on top of large image classification models, capable of representing thousands of semantic classes from common and rare objects to rooms and regions.

- A simple, yet effective, method to localize in 3D, the semantic classes from the text-based map.

- A method for grounded LLM spatial reasoning over the text-based map to generate plans from given user task specifications.

Through quantitative experiments, we demonstrate that the localizations from our map perform comparably in precision and recall against state-of-the-art open vocabulary maps while storing orders of magnitudes less information. Real-world robot experiments demonstrate the effectiveness of our map at grounding a LLMs to reason from task descriptions and generate feasible navigation plans.

## 2 Related Work

**Semantic Map Representations.** A semantic map represents the scene geometry and assigns semantic classes to that geometry. Earlier works leveraged 2D or 3D segmentation and object detection models trained on a fixed set of classes to generate class annotations which are stored in the map [15, 16, 17, 18, 19]. Semantic maps can be extended into scene graphs by further annotating relationships between labeled instances [20, 21, 22], such as objects next to each other or objects contained in a room. Recent interest has been directed towards open vocabulary representations where rather than explicit classes, the map is annotated with embedding vectors queryable by arbitrary text. Such maps have attached embeddings to dense scene geometry [23, 24, 13], to segmented instances or objects [25, 26, 14], as well as into scene graphs [27, 28, 29].

**Object Goal Navigation.** While some works in ObjectNav applied end-to-end learning [30, 31, 32, 33, 34, 35] or explored semantics-agnostic exploration strategies [36], many works have opted to maintain a map during runtime. Storing what has already been observed in a map can help in picking semantically relevant locations to explore further [37, 38], as a more privileged input to a learned policy [3, 35], to explore unseen frontiers [39, 40, 41, 42, 43], and to reduce exploration when given followup goals [5]. Multiple works have applied LLMs to reason over the map semantics. Given a target object, commonsense reasoning from an LLM is useful for suggesting related objects and rooms for more efficient selection of regions, frontiers, or objects to explore [40, 42, 43, 44]. LLMs have also been applied to translate natural language user task specifications into actionable navigation goals within the map [13, 14, 5].

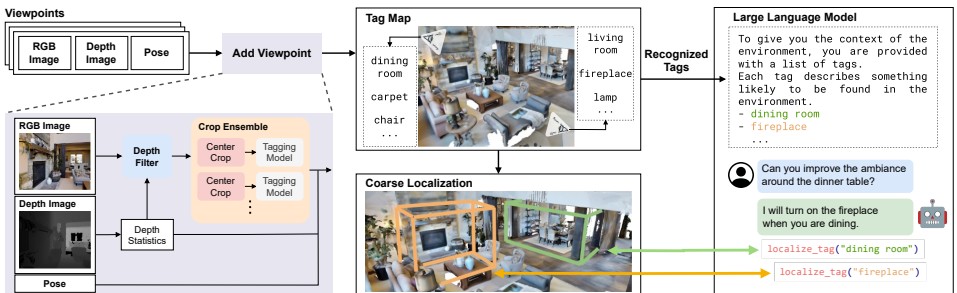

Figure 2: Our text-based map (*tag map*) stores viewpoints from a scene and the entities (tags) recognized by an image tagging model from these viewpoints. The stored tags are used to create a context prompt for grounding an LLM to generate navigation plans over the scene. Tags in the map can be coarsely localized in space and fed into the LLM for further spatial reasoning.

**Multi-Label Classification.** A well-studied computer vision task [45, 46], also known as image tagging, where the objective is to recognize the semantic classes contained within an image. In contrast to segmentation or detection, tagging does not require localization of the classes within the image. Tagging models are a mature technology with several commercially available APIs [47, 48, 49] that can recognize thousands of classes. In this work, we use the Recognize Anything (RAM) family of open-source models [50, 51] on par with most commercially available APIs.

# 3 Text-Based Map Representation

The core of our approach is the text-based tag map which stores the unique entities (tags) recognized by an image tagging model and relates them to the viewpoints they were recognized from. Its data structure can be implemented as a hash table, where the keys are the unique recognized entities and the values are references to the corresponding viewpoints. Each viewpoint stores a unique ID, its camera pose, and a depth statistic used later for localization. We emphasize that the viewpoint RGB and depth images are not stored in the map. An overview of our framework is shown in Fig. 2.

## 3.1 Map Construction

Building a tag map from RGB-D images and poses of a set of viewpoints is shown in Fig. 2. From the depth image, we compute the mean and median depth, along with the 80th quantile depth value representing the far plane distance of the viewpoint frustum. A filtering step checks the mean and median depth and discards the frame if it is considered a close-up view. The RGB image is then forwarded through a tagging model ensemble, where each ensemble member receives a different centered cropped version of the image that removes a small portion of the edges. The final tags are the tags agreed on by the ensemble, filtering out false positive tags due to poorly observed objects around the edges of an image. Effects of the depth and crop ensemble filters are visualized in Fig. 3. The tags, poses, and viewpoint frustum distances are registered in the tag map.

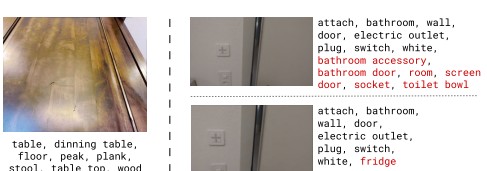

Figure 3: **Left**: Example of a close-up view discarded by the depth filtering step. Such images tend to generate inaccurate tags due to a lack of context. **Right**: An image crop results in significant differences in the tags recognized by an image tagging model. Using a crop augmented ensemble of tagging models filters out the inconsistent tags in red.

## 3.2 Coarse-Grained Localization

Given a tag, multi-view consistency over the tag's corresponding viewpoints is used to generate localized regions in the tag's near vicinity, as demonstrated in Fig. 4. The procedure is similar to Space Carving [52], where viewpoint intersections are used to reconstruct 3D geometry. Unlike Space Carving, viewpoints in the tag map may be generated from partial views of an entity.

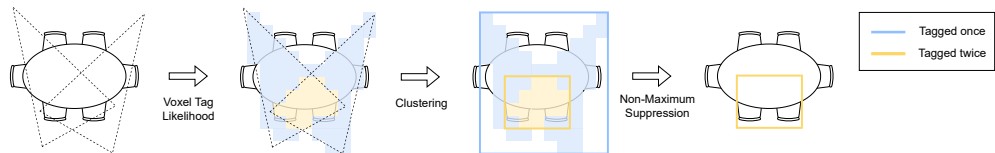

Figure 4: Illustrative coarse localization example of "table" with two viewpoints. Spaces covered by the viewpoints are voxelized and each voxel is assigned votes based on the number of viewpoints that contain it. Clustering voxels of at least $v$ votes generates proposals at a confidence level of $v$. Non-maximum suppression is applied to remove redundant proposals.

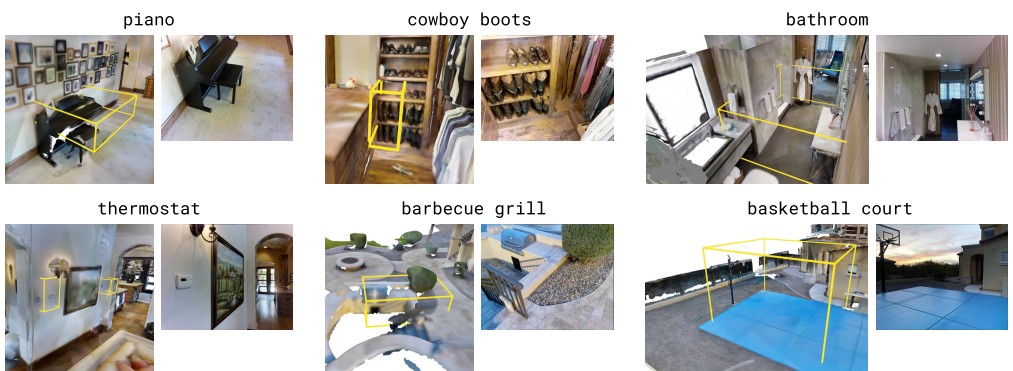

Figure 5: Example coarse localizations for various objects and locations, along with images where they were recognized. The large vocabulary of image tagging models allows our framework to localize uncommon categories such as "thermostat", "cowboy boots" and "basketball court".

Localizing a tag begins with retrieving that tag's corresponding viewpoints and mapping each viewpoint to its 3D frustum using the stored far plane distance. The space covered by the viewpoints is voxelized and each voxel is assigned votes corresponding to the number of viewpoints that contains it. The voxel votes represent a likelihood of the tag over the space. Localization proposals at a confidence level $v$ are extracted by thresholding out voxels below $v$ votes and clustering the remaining voxels using DBSCAN [53]. A confidence level of $n$ indicates that a proposal is consistent across at least $n$ viewpoints. We set a series of clustering thresholds to capture not just the global but also local maxima of the voxel likelihood. Lastly, a non-maximum suppression step is applied to remove proposals that contain within it proposals of higher confidence levels. Fig. 5 shows example coarse localizations for various tags of objects and locations.

### 3.3 Grounding Large Language Models

We opt for the simple method of grounding the LLM by appending the list of unique tags from the tag map into the prompt of the LLM, as shown in Figure 2. The LLM is additionally given access to an API for requesting information from the tag map. The API exposes the following functions:

- `localize_tag(tag)`: Returns the proposals for the tag along with their confidence levels.
- `region_region_dist(r1, r2)`: Computes the distance between regions `r1` and `r2`.
- `point_region_dist(p, r)`: Computes the distance to reach region `r` from point `p`.

The API enables the LLM to reason spatially about the tags in its prompt. For example, given a query of `"go to the kitchen fridge"`, the LLM can localize "kitchen" and "fridge" and set the navigation goal to be the fridge proposal with a small distance to a kitchen proposal.

## 4 Evaluating Coarse-Grained Localizations

The coarse localizations produced by our framework generally do not precisely contain the localized entity as seen in Fig. 5. However, such localizations can still be useful navigation goals for reaching the entities. Evaluating coarse localizations using metrics such as Intersection Over Union (IoU)

fails to assess localizations in terms of "imprecise but still useful" navigation goals. For example, localizing a small area within a bedroom would be a useful goal for navigating to the bedroom but have poor IoU against the room's bounding box label.

## 4.1 Metrics for Evaluating Coarse-Grained Localizations

Our goal is to produce a proposal $P$ that is useful for locating the desired entity. In other words, if the robot reaches the region $P$, it should be close enough to an instance of the desired entity $E \in \mathcal{E}$. This is aligned with the objective of the Habitat ObjectNav Challenge [2] benchmark, where an agent is successful in navigating to a goal object if the distance between the agent and the nearest goal object is below 1.0 m[1] and the object can be viewed with basic yaw and pitch movements of the camera. We quantify the usefulness of $P$ as the expected length of the shortest path an agent must travel from $P$ to reach *any* desired entity instance in the set $\mathcal{E}$. We denote this quantity as the Proposal to Entities Distance (P2E) defined as

$$\text{P2E} = \mathop{\mathbb{E}}_{p \sim P} \left[ \min_{E \in \mathcal{E}} d(p, E) \right], \tag{1}$$

where $d(\cdot, \cdot)$ is the shortest path length function. We can evaluate P2E over a set of proposals and classify proposals with P2E under a set threshold (e.g. 1.0 m) as relevant. We refer to the portion of relevant proposals for that threshold as the Precision at Threshold and use it to measure the *precision* of that set of proposals.

Symmetrically, we define the Entity to Proposals Distance (E2P) which evaluates for an entity instance $E$ the shortest expected path length from $E$ to reach any proposal for that entity. For a set of entity instances, we compute the portion with E2P below a threshold for a given set of proposals as a measure of the *recall* for that set of proposals, which we denote as the Recall at Threshold.

The expected shortest path length is difficult to compute exactly so we approximate it by constructing a grid graph over the scene that avoids collisions with the scene geometry, allowing the computation of approximate shortest paths. Nodes of the grid graph are assigned to the proposals and entity instances and the expectation is approximated by averaging over the assigned nodes. Computing the approximations of P2E and E2P is equivalent to computing the Directed Average Hausdorff Distance (DAHD) [54], a metric commonly used in medical image segmentation.

## 4.2 Evaluation Implementation

Evaluations are performed on the Matterport3D dataset [55]. The evaluation is separated into object and room/region evaluations. Object evaluation is done on the 21 common object classes defined by the Habitat ObjectNav Challenge [2]. For regions, we evaluate all classes in Matterport3D except for "other room", "junk", "no label", "outdoor", "entryway/foyer/lobby", and "dining booth". These classes were ignored as they did not refer to actual rooms or corresponded to ambiguously labeled regions. When evaluating a tag map, each class label is mapped to a list of corresponding tags. Proposals are generated for all the corresponding tags that exist in the tag map and are grouped to form the set of proposals for the class. Unless stated otherwise, we report the precision and recall over all classes by averaging the precision and recall across classes. Further details on the implementation of the evaluation are found in Appendix D.

# 5 Results

## 5.1 Comparison with Embedding Based Maps

Tag map localizations are compared against localizations from open vocabulary embedding-based maps using the proposed precision and recall metrics. We chose to compare against OpenScene [23] for 3D segmentation using a point cloud representation with an embedding for each point, and OpenMask3D [25] for 3D instance segmentation which stores 3D instance masks with embeddings

---

[1]In the challenge, another 0.1m is allowed, thus the distance threshold is actually 1.1 m.

for each mask. We query all embeddings with all class labels at once, then assign each embedding a class based on the best matching class following Peng et al. [23]. Another option is to query classes individually and assign embeddings a class if they match above a threshold as done in Lu et al. [26]. However, picking a suitable threshold is difficult and thresholds may not transfer across scenes. For OpenScene, we cluster the segmentation output to obtain instance bounding boxes following Lu et al. [26]. The comparison is only done over the Matterport3D test split as OpenScene uses the train and validation splits during training.

| Object Precision-Recall | Precision at Threshold [m] | | | | Recall at Threshold [m] | | | |
|---|---|---|---|---|---|---|---|---|
| | 0.1 | 0.5 | 1.0 | 2.0 | 0.1 | 0.5 | 1.0 | 2.0 |
| OpenScene [23] | **0.31** | 0.36 | 0.41 | 0.46 | 0.12 | 0.29 | 0.61 | **0.88** |
| OpenMask3D [25] | 0.29 | 0.35 | 0.41 | 0.48 | 0.05 | 0.19 | 0.41 | 0.60 |
| Tag Map (Ours) | 0.28 | **0.39** | **0.46** | **0.53** | **0.13** | **0.42** | **0.65** | 0.82 |

Table 1: Evaluation on object classes against open-vocabulary map representations.

| Region Precision-Recall | Precision at Threshold [m] | | | | Recall at Threshold [m] | | | |
|---|---|---|---|---|---|---|---|---|
| | 0.5 | 1.0 | 2.0 | 3.0 | 0.5 | 1.0 | 2.0 | 3.0 |
| OpenScene [23] | 0.32 | 0.33 | 0.36 | 0.38 | **0.22** | **0.36** | **0.56** | **0.64** |
| OpenMask3D [25] | **0.56** | **0.58** | **0.61** | **0.63** | 0.10 | 0.26 | 0.49 | 0.63 |
| Tag Map (Ours) | 0.46 | 0.50 | 0.54 | 0.56 | 0.11 | 0.22 | 0.44 | **0.64** |

Table 2: Evaluation on region classes against open-vocabulary map representations.

The comparison results are reported in Tables 1 and 2. For objects, the tag map outperforms both embedding-based maps for most precision and recall thresholds. However, the embedding-based maps have better precision when the threshold is set to a smaller value (e.g. 0.1 m), due to these methods storing geometric information to produce tighter instance bounding boxes. For regions, OpenMask3D achieves the best precision across thresholds as it produces smaller instance proposals that are often precisely located within the region label. However, these smaller proposals have lower recall as the larger size of region labels results in higher E2P. Conversely, OpenScene tends to produce much larger proposals for region classes resulting in lower E2P and better recall. The tag map achieves good precision for region classes but the worst recall compared to the other methods. Note that larger precision-recall thresholds are used for the regions as region labels are much larger than object labels. We also compared the memory usage of the maps in Fig. 6. The tag map uses much less memory than the embedding-based maps as storing the embedding vectors requires significantly more memory than text-based tags. The comparison is especially stark against OpenScene which stores per-point embeddings for a scene point cloud.

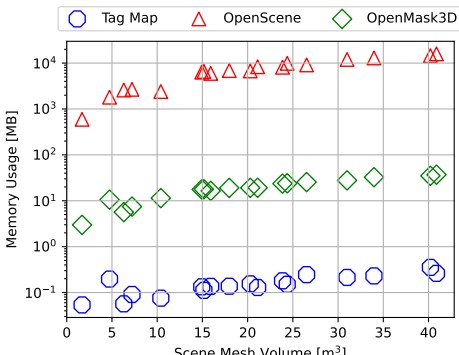

Figure 6: Memory use of the tag map and open-vocabulary maps across scenes of various sizes. Scene size is measured as the voxel volume after voxeling the scene mesh

## 5.2 Comparison with CLIP Viewpoint Retrieval
The recent work of Chang et al. [5] proposed to use store viewpoint CLIP embeddings and to search over the embeddings when given an object query. We modify our coarse localization method to retrieve viewpoints using CLIP instead of using the relevant tags. To make the comparison fair, for both methods, we retrieve only the top $K$ most confident viewpoints. For tags, we use the tag prediction confidences from the tagging model. We also include the unmodified tag localization method as a baseline, denoted as "all views". Precision and recall comparisons for the two methods at different thresholds are reported in Fig. 7. Retrieval via tags consistently outperforms CLIP

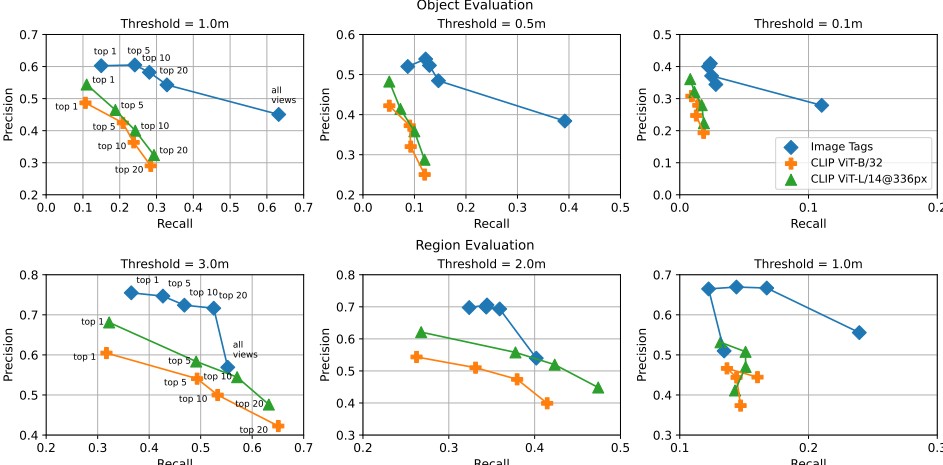

Figure 7: Comparing the effect of retrieving viewpoints using tags against using CLIP embeddings on coarse localization precision and recall. We compare retrieving the top {1, 5, 10, 20} most confident viewpoints for each method. For using tags, we also include "all views", which retrieves all tagged viewpoints. Using tags consistently outperforms CLIP embeddings in precision for both object and region classes as well as consistently better recall for objects.

retrieval in terms of precision for the same $K$, suggesting that the performance of image embeddings is behind that of models specialized in recognition. For object classes, tag-based retrieval also outperforms CLIP in terms of recall. We observe a general trend of trading off precision for recall for increasing $K$ which is to be expected. For object classes, retrieving all relevant viewpoints ("all views") results in a significant increase in recall while this only produces a slight increase in recall for region classes. This is because a single scene only contains a few large region instances but many smaller object instances. Therefore, only a few views are necessary to recall all region instances.

## 5.3 Ablation of the Map Construction Modules

| Object Precision-Recall | Precision at Threshold [m] | | | | Recall at Threshold [m] | | | |
|---|---|---|---|---|---|---|---|---|
| | 0.1 | 0.5 | 1.0 | 2.0 | 0.1 | 0.5 | 1.0 | 2.0 |
| RAM | **0.28** | **0.38** | **0.45** | **0.52** | 0.11 | 0.39 | 0.63 | 0.82 |
| RAM w/o ensemble | 0.23 | 0.33 | 0.39 | 0.45 | **0.13** | **0.43** | **0.67** | **0.85** |
| RAM w/o depth | 0.27 | **0.38** | 0.44 | 0.51 | 0.11 | 0.39 | 0.64 | 0.82 |
| RAM++ | 0.26 | 0.36 | 0.43 | 0.49 | 0.12 | 0.40 | 0.64 | 0.83 |

Table 3: Tag map construction modules ablations evaluated on precision and recall for object classes. For conciseness, crop ensemble filtering is referred to as "ensemble" and depth filtering as "depth".

We ablate the tagging model, the crop ensemble filter, and the depth filter. We compared the tagging models RAM [50] and its successor RAM++ [51], finding that RAM++ did not outperform RAM in our evaluations. This is likely because RAM++ is designed to improve the recognition of rare classes and our evaluation is limited to more common classes from Matterport3D. The depth filter was found to have an insignificant impact on the evaluation performance. We believe this is because the depth filtering removes uninformative views which often generate incorrect tags that are generally unrelated to the classes considered in the evaluation (see Fig. 3). Removing the crop ensemble filter results in a notable drop in precision with a smaller gain in recall, suggesting that the crop ensemble filter provides a favorable tradeoff between filtering false positive tags and over-filtering true positives. Ablation results over object classes are reported in Table 3 while region class results are reported in Appendix A.1.

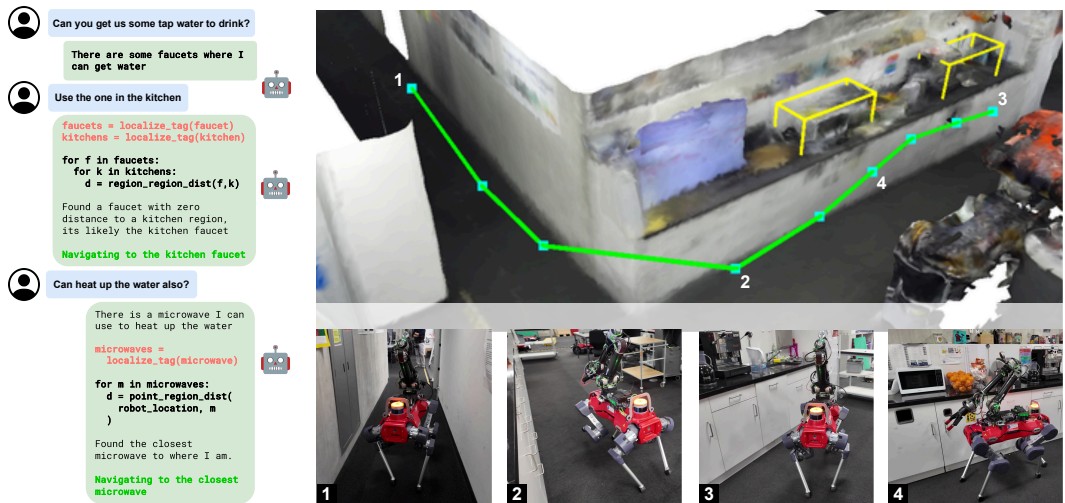

Figure 8: An example of grounded navigation using a tag map on a real robot. A user asks the system to assist with some task through a chat interface. First, the LLM uses the tag map context to propose relevant entities for the task. Next, the LLM queries the tag map API for additional information on the entities of interest which it then uses to reason spatially and produce a navigation plan.

## 5.4 Experiments of Grounded Navigation on the Real Robot

Integration of the tag map with an LLM for grounded navigation was demonstrated on the legged robot ANYmal [56]. A dataset from a lab/office scene was collected and viewpoint poses were computed using COLMAP [57, 58] to construct the tag map and a scene mesh. The mesh is used for visualization and to build a pose graph connecting neighboring viewpoints in the tag map that have a traversable path to each other. The pose graph produces global path plans that are tracked by a local planner based on Cao et al. [59]. We used GPT-4 [60] as the LLM and leveraged its function calling capabilities to integrate it with the tag map API. The LLM is allowed to make multiple function calls before giving its response. This allows the LLM to, for example, localize relevant tags using `localize_tag()` and then reason about them spatially using `region_region_distance()` in a single response. This can be thought of as the LLM generating scene graph annotations between the localized proposals. We also prompt the LLM to consider the localization confidence levels such that it will prefer selecting more confident localizations to navigate to.

A user is given a chat interface for conversing with the LLM which allows the LLM to ask the user for additional clarifying information and for the user to provide follow-up tasks building on top of previous queries. Fig. 8 demonstrates examples of grounded navigation addressing tasks given by a user. We focus on tasks where the relevant entities for solving the task are not explicitly mentioned and must be inferred by the LLM through the tag map context. In addition to the robot demonstrations, a more thorough evaluation of the Tag Map grounded navigation pipeline over a set of test user queries is presented in Appendix F.

## 6 Conclusions, Limitations, and Future Work

This work presented a text-based map that can represent an extensive set of entities, including rare objects and regions, and coarsely localize them in 3D. Being text-based allowed the map to seamlessly ground an LLM through its prompt. Interfacing the LLM's function calling with the map enabled further spatial reasoning by the LLM on the localized entities. The tagging model used in this work was prone to producing false positive tags which risked polluting the context given to the LLM. However, the false positive tags were often spurious concepts having little to do with the more practical tasks we tested and were mostly ignored by the LLM. Though sometimes the LLM generated infeasible plans due to such tags. For the evaluations conducted on Matterport3D, we were limited to the viewpoints available in the dataset. It remains an open question if different viewpoint collection strategies can improve the map's localizations. Additionally, we only considered static scenes and did not consider cases where entities could have moved, leaving this for future work.

**Acknowledgments**

This work was supported by Huawei Tech R&D (UK) through a research funding agreement, by the Swiss National Science Foundation through the National Centre of Competence in Digital Fabrication (NCCR dfab), and by an ETH RobotX research grant funded through the ETH Zurich Foundation. Moreover, this work has been conducted as part of ANYmal Research, a community to advance legged robotics. We would also like to thank Jie Tan for helpful discussions on the paper.

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

# Appendix

## A   Additional Results

### A.1   Ablation Results of the Map Construction Modules for Region Classes

| Region Precision-Recall for | Precision at Threshold [m] | | | | Recall at Threshold [m] | | | |
|---|---|---|---|---|---|---|---|---|
| | 0.5 | 1.0 | 2.0 | 3.0 | 0.5 | 1.0 | 2.0 | 3.0 |
| RAM | **0.49** | **0.51** | **0.54** | **0.57** | **0.06** | 0.13 | 0.40 | 0.55 |
| RAM w/o ensemble | 0.46 | 0.48 | 0.51 | 0.53 | **0.06** | 0.16 | **0.42** | **0.57** |
| RAM w/o depth | 0.48 | **0.51** | **0.54** | 0.56 | **0.06** | **0.19** | **0.42** | 0.56 |
| RAM++ | 0.44 | 0.46 | 0.49 | 0.53 | **0.06** | 0.15 | **0.42** | **0.57** |

Table 4: Tag map ablations evaluated on precision and recall for region classes. For conciseness, crop ensemble filtering is referred to as "ensemble" and depth filtering as "depth".

### A.2   Tag Map Precision and Recall for Each Class in Matterport3D

| Object Precision-Recall | Precision at Threshold [m] | | | | Recall at Threshold [m] | | | |
|---|---|---|---|---|---|---|---|---|
| | 0.1 | 0.5 | 1.0 | 2.0 | 0.1 | 0.5 | 1.0 | 2.0 |
| bathtub | 0.23 | 0.28 | 0.30 | 0.34 | 0.07 | 0.24 | 0.60 | 0.88 |
| bed | 0.53 | 0.61 | 0.65 | 0.69 | 0.05 | 0.41 | 0.68 | 0.94 |
| cabinet | 0.25 | 0.34 | 0.41 | 0.49 | 0.11 | 0.45 | 0.71 | 0.89 |
| chair | 0.39 | 0.53 | 0.62 | 0.72 | 0.18 | 0.44 | 0.63 | 0.79 |
| chest_of_drawers | 0.10 | 0.17 | 0.22 | 0.28 | 0.13 | 0.44 | 0.69 | 0.86 |
| clothes | 0.09 | 0.13 | 0.15 | 0.18 | 0.07 | 0.52 | 0.77 | 0.90 |
| counter | 0.34 | 0.41 | 0.46 | 0.52 | 0.11 | 0.36 | 0.62 | 0.81 |
| cushion | 0.21 | 0.40 | 0.51 | 0.58 | 0.09 | 0.36 | 0.59 | 0.82 |
| fireplace | 0.36 | 0.50 | 0.53 | 0.59 | 0.09 | 0.35 | 0.66 | 0.88 |
| gym_equipment | 0.49 | 0.60 | 0.71 | 0.78 | 0.06 | 0.16 | 0.33 | 0.65 |
| picture | 0.42 | 0.58 | 0.68 | 0.78 | 0.20 | 0.49 | 0.69 | 0.83 |
| plant | 0.26 | 0.36 | 0.43 | 0.51 | 0.13 | 0.43 | 0.64 | 0.82 |
| seating | 0.24 | 0.32 | 0.37 | 0.45 | 0.03 | 0.23 | 0.35 | 0.51 |
| shower | 0.21 | 0.29 | 0.35 | 0.43 | 0.10 | 0.40 | 0.72 | 0.90 |
| sink | 0.18 | 0.33 | 0.43 | 0.51 | 0.17 | 0.55 | 0.81 | 0.92 |
| sofa | 0.30 | 0.42 | 0.50 | 0.56 | 0.07 | 0.31 | 0.50 | 0.72 |
| stool | 0.06 | 0.09 | 0.12 | 0.17 | 0.15 | 0.42 | 0.58 | 0.76 |
| table | 0.30 | 0.44 | 0.51 | 0.61 | 0.19 | 0.53 | 0.73 | 0.88 |
| toilet | 0.45 | 0.55 | 0.59 | 0.67 | 0.08 | 0.44 | 0.74 | 0.91 |
| towel | 0.29 | 0.41 | 0.52 | 0.59 | 0.03 | 0.25 | 0.51 | 0.67 |
| tv_monitor | 0.16 | 0.31 | 0.39 | 0.49 | 0.17 | 0.46 | 0.73 | 0.86 |

Table 5: Precision-recall for common object classes evaluated over all 90 scenes of Matterport3D

| Region Precision-Recall | Precision at Threshold [m] | | | | Recall at Threshold [m] | | | |
|---|---|---|---|---|---|---|---|---|
| | 0.5 | 1.0 | 2.0 | 3.0 | 0.5 | 1.0 | 2.0 | 3.0 |
| balcony | 0.26 | 0.31 | 0.38 | 0.43 | 0.00 | 0.00 | 0.14 | 0.45 |
| bar | 0.00 | 0.00 | 0.00 | 0.00 | 0.33 | 0.33 | 0.33 | 0.33 |
| bathroom | 0.30 | 0.33 | 0.35 | 0.38 | 0.21 | 0.38 | 0.68 | 0.86 |
| bedroom | 0.47 | 0.49 | 0.52 | 0.54 | 0.06 | 0.16 | 0.56 | 0.85 |
| classroom | 0.60 | 0.60 | 0.60 | 0.60 | 0.00 | 0.00 | 0.50 | 0.50 |
| closet | 0.08 | 0.09 | 0.13 | 0.16 | 0.16 | 0.23 | 0.48 | 0.63 |
| dining room | 0.47 | 0.50 | 0.53 | 0.56 | 0.03 | 0.16 | 0.60 | 0.81 |
| garage | 0.69 | 0.72 | 0.72 | 0.77 | 0.00 | 0.00 | 0.29 | 0.57 |
| hallway | 0.54 | 0.59 | 0.68 | 0.73 | 0.05 | 0.14 | 0.43 | 0.67 |
| kitchen | 0.38 | 0.40 | 0.42 | 0.45 | 0.09 | 0.29 | 0.69 | 0.88 |
| laundryroom/mudroom | 0.68 | 0.70 | 0.72 | 0.72 | 0.00 | 0.17 | 0.63 | 0.89 |
| library | 1.00 | 1.00 | 1.00 | 1.00 | 0.00 | 0.00 | 0.50 | 0.50 |
| living room | 0.40 | 0.42 | 0.47 | 0.51 | 0.07 | 0.19 | 0.43 | 0.61 |
| meetingroom/conferenceroom | 0.32 | 0.36 | 0.36 | 0.44 | 0.04 | 0.08 | 0.12 | 0.28 |
| office | 0.36 | 0.38 | 0.44 | 0.48 | 0.06 | 0.14 | 0.37 | 0.43 |
| porch/terrace/deck/driveway | 0.54 | 0.55 | 0.58 | 0.61 | 0.05 | 0.11 | 0.36 | 0.52 |
| rec/game | 1.00 | 1.00 | 1.00 | 1.00 | 0.00 | 0.00 | 0.06 | 0.06 |
| spa/sauna | 0.34 | 0.37 | 0.37 | 0.40 | 0.02 | 0.07 | 0.16 | 0.23 |
| stairs | 0.26 | 0.29 | 0.35 | 0.38 | 0.08 | 0.20 | 0.53 | 0.72 |
| tv | 0.71 | 0.71 | 0.82 | 0.88 | 0.00 | 0.31 | 0.62 | 0.62 |
| utilityroom/toolroom | 0.00 | 0.00 | 0.00 | 0.00 | 0.00 | 0.00 | 0.00 | 0.00 |
| workout/gym/exercise | 0.89 | 0.89 | 0.89 | 0.92 | 0.00 | 0.00 | 0.38 | 0.75 |

Table 6: Precision-recall for region classes evaluated over all 90 scenes of Matterport3D

# B  Tag Map Parameters

The map construction and localization parameters used for all experiments are presented in Table 7 and 8 respectively.

| Parameter | Value |
|---|---|
| Tagging model | ram_swin_large [50] |
| Crop ensemble border crop percentages | [5%, 10%] |
| Depth filter mean threshold | 0.6 m |
| Depth filter median threshold | 0.6 m |

Table 7: Tag map construction parameters

| Parameter | Value |
|---|---|
| Viewpoint near plane distance | 0.2 m |
| Viewpoint far plane distance | $80^{th}$ percentile of depth values |
| Voxel size | 0.2 m |
| DBSCAN eps | 0.4 m |
| DBSCAN minimum points | 5 |
| Normalized votes clustering thresholds | [0.00, 0.25, 0.50, 0.75] |

Table 8: Tag map localization parameters

Note that the clustering thresholds on the voxel votes are set relative to the normalized votes, i.e. the voxel votes normalized by the maximum votes of any voxel.

## C   LLM Chat Prompt for Real-Robot Experiments

```
You are a helpful robot assistant.

To give you some knowledge of your environment, you are provided with
a set of tags.  Each tag describes something that's likely to be found
in the environment.  You can use these tags to help you with assisting
the user.

Note that the tags are not perfect.  Some tags may be incorrect and
the tags do not cover everything in the environment.  Therefore, try to
pick the tag which you are most confident in.

You can query for the regions in the environment where a tag is
localized and also get the confidence level for each region.

When making a statement about a tag, do not say that the tag is
definitely in the environment, instead reference the confidence level.

If there are no tags directly related to the user's request, suggest
tags which may be related to the user's request and ask the user if
they would like to know more about the suggested tags.

List of tags in the format '[id] - [tag]'
0 - <tag 0>
1 - <tag 1>
    .
    .
    .
N - <tag N>
```

# D  Details of the Evaluation Implementation

## D.1  Grid Graph Construction

Computing the P2E and E2P requires solving for the shortest paths within a scene, which we compute approximately using a 3D grid graph that spans the scene without collision with the scene geometry. Note that the grid graph connects nodes without consideration of traversability. For example, the free space above a table would be connected but would not be traversable by a wheeled robot. An overview of the steps to construct a grid graph is given in Algorithm 1.

---

**Algorithm 1** Scene Grid Graph Generation

---

1: **Input:** Scene Mesh $\mathcal{M}$
2: **Output:** Set of nodes $\mathcal{N}$ and edges $\mathcal{E}$
3: Determine the min and max bounds of $\mathcal{M}$
4: Generate a set of nodes $\mathcal{N}$ following a grid spanning these bounds
5: **for** each node $n \in \mathcal{N}$ **do**
6:     **if** InsideScene$(n, \mathcal{M})$ is **false then**
7:         Remove $n$ from $\mathcal{N}$
8:     **end if**
9: **end for**
10: Initialize empty set of edges $\mathcal{E}$
11: **for** each node $n \in \mathcal{N}$ **do**
12:     **for** each immediate neighbor $n'$ of $n$ in the grid **do**
13:         **if** $(n, n') \notin \mathcal{E}$ and CollisionFree$(n, n', \mathcal{M})$ is **true then**
14:             Add edge $(n, n')$ to $\mathcal{E}$
15:         **end if**
16:     **end for**
17: **end for**
18: **Return** $\mathcal{N}, \mathcal{E}$

---

The grid graphs are generated at a grid resolution of 0.5 m. We check if a point is inside the scene by checking its nearest neighboring vertices. First, if the mean distance to these vertices is greater than a threshold of 2 m we consider the point outside the scene. Second, we use the normals of each neighboring vertex to check if the point is locally inside the mesh at that vertex by computing the dot product of the vertex-to-point vector and the normal. We average the dot products across all neighboring vertices and consider the point outside the mesh if the average exceeds a threshold of 0. To evaluate if two points are collision-free, we cast a ray from one to the other and check for collisions against the scene mesh.

## D.2  Assigning Grid Graph Nodes

The assignment of grid graph nodes to instance proposals is given in Algorithm 2. For assigning nodes to labeled entity instances, the algorithm differs depending on whether the entity is an object or a region, as outlined in Algorithms 3 and 4 respectively.

The node assignment algorithms differ in the use of inflation to assign additional nodes. In the case of instance proposals, inflation is only used when the proposal does not contain any nodes. Here we set the inflation amount $\delta$ such that the extent of the inflated proposal is at least larger than the space between nodes in the grid graph. For object labels, inflation is always used to assign additional nodes. In this case, we set $\delta$ to capture nodes 1 m away from the label bounding box, following the success criteria of the Habitat ObjectNav Challenge [2]. Lastly, no inflation is used for region labels as they are large enough to contain at least some nodes.

If no nodes can be assigned to an instance proposal or labeled entity, then the P2E or E2P cannot be computed. In these cases, we ignore that instance proposal or labeled entity when later computing the precision and recall metrics.

**Algorithm 2** Assign Grid Graph Nodes to an Instance Proposal

---

1: **Input:** Set of nodes $\mathcal{N}$, Instance proposal bounding box $P$, Scene mesh $\mathcal{M}$, Inflation amount $\delta$
2: **Output:** Set of nodes $\mathcal{N}_{\text{result}}$
3: Initialize $\mathcal{N}_{\text{result}} \leftarrow \emptyset$
4: Find nodes $\mathcal{N}_{\text{in}} \subseteq \mathcal{N}$ that are contained within $P$
5: **if** $\mathcal{N}_{\text{in}} \neq \emptyset$ **then**
6:     **return** $\mathcal{N}_{\text{in}}$
7: **end if**
8: Inflate $P$ by a fixed amount $\delta$ to obtain a new bounding box $P'$
9: Find nodes $\mathcal{N}_{\text{in}} \subseteq \mathcal{N}$ that are contained within $P'$
10: **for** each node $n \in \mathcal{N}_{\text{in}}$ **do**
11:     Find the nearest point $p \in P$ to $n$
12:     **if** CollisionFree$(n, p, \mathcal{M})$ is **true then**
13:         Add $n$ to $\mathcal{N}_{\text{result}}$
14:     **end if**
15: **end for**
16: **return** $\mathcal{N}_{\text{result}}$

---

**Algorithm 3** Assign Grid Graph Nodes to a Labeled Object Instance

---

1: **Input:** Set of nodes $\mathcal{N}$, Object label bounding box $O$, Scene mesh $\mathcal{M}$, Inflation amount $\delta$
2: **Output:** Set of nodes $\mathcal{N}_{\text{result}}$
3: Find nodes $\mathcal{N}_{\text{in}} \subseteq \mathcal{N}$ that are contained within $O$
4: $\mathcal{N}_{\text{result}} \leftarrow \mathcal{N}_{\text{in}}$
5: Inflate $O$ by a fixed amount $\delta$ to obtain a new bounding box $O'$
6: Find additional nodes $\mathcal{N}'_{\text{in}} \subseteq \mathcal{N}$ that are contained within $O' \setminus O$
7: **for** each node $n \in \mathcal{N}'_{\text{in}}$ **do**
8:     Find the nearest point $p \in O$ to $n$
9:     **if** CollisionFree$(n, p, \mathcal{M})$ is **true then**
10:         Add $n$ to $\mathcal{N}_{\text{result}}$
11:     **end if**
12: **end for**
13: **return** $\mathcal{N}_{\text{result}}$

---

**Algorithm 4** Assign Grid Graph Nodes to a Labeled Region Instance

---

1: **Input:** Set of nodes $\mathcal{N}$, Region label bounding box $R$, Scene mesh $\mathcal{M}$
2: **Output:** Set of nodes $\mathcal{N}_{\text{result}}$
3: Find nodes $\mathcal{N}_{\text{in}} \subseteq \mathcal{N}$ that are contained within $R$
4: $\mathcal{N}_{\text{result}} \leftarrow \mathcal{N}_{\text{in}}$
5: **return** $\mathcal{N}_{\text{result}}$

---

### D.3 Mapping Object Class Labels to Tags

Matterport3D includes a raw class label for all labeled objects. For example the raw labels "arm chair" and "bean bag chair" are grouped into the common object class of "chair". Both "arm chair" and "bean bag chair" are within the set of classes able to be recognized by the tagging model. Because of this, for each common object class, we identify the corresponding raw classes and map them to matching tags within the vocabulary of the tagging model. We also manually identified tags that corresponded to a common object class and included them in the mapping. The mapping for the common object classes to tags are defined as follows:

**bathtub**: bath, jacuzzi

**bed**: bed, bed frame, bunk bed, canopy bed, cat bed, dog bed, futon, hammock, headboard, hospital bed, infant bed, mattress

**cabinet**: armoire, bathroom cabinet, cabinet, cabinetry, closet, file cabinet, kitchen cabinet, medicine cabinet, side cabinet, tv cabinet, wine cabinet

**chair**: armchair, beach chair, bean bag chair, beanbag, chair, computer chair, feeding chair, folding chair, office chair, rocking chair, swivel chair, throne

**chest_of_drawers**: bureau, drawer, dresser, nightstand

**clothes**: baby clothe, baseball hat, bathrobe, bathroom accessory, bikini, bikini top, blouse, christmas hat, cloak, clothing, coat, cocktail dress, corset, costume, cowboy hat, crop top, denim jacket, dress, dress hat, dress shirt, dress shoe, dress suit, evening dress, fur coat, gown, halter top, hat, headdress, headscarf, hoodie, jacket, jeans, jockey cap, kilt, kimono, lab coat, lace dress, laundry, leather jacket, maxi dress, miniskirt, overcoat, pants, pantyhose, polo neck, polo shirt, raincoat, robe, safety vest, scarf, shirt, ski jacket, sports coat, straw hat, sun hat, suspenders, sweat pant, sweater, sweatshirt, t shirt, t-shirt, trench coat, underclothes, vest, visor, waterproof jacket, wedding dress, wrap dress

**counter**: bar, buffet, counter, counter top, island, kitchen counter, kitchen island, wet bar

**cushion**: pillow, throw pillow

**fireplace**: fireplace, mantle

**gym_equipment**: barbell, dumbbell, stationary bicycle, training bench, treadmill, weight

**picture**: art, art print, couple photo, decorative picture, drawing, family photo, group photo, movie poster, oil painting, photo, photo frame, picture, picture frame, portrait, poster, publicity portrait, reflection, wedding photo

**plant**: bush, flower, grass, houseplant, plant, tree

**seating**: bench, church bench, park bench, seat, window seat

**shower**: shower, shower door, shower head

**sink**: basin, bathroom sink, sink

**sofa**: couch, loveseat

**stool**: bar stool, footrest, music stool, step stool, stool

**table**: altar, billiard table, changing table, cocktail table, computer desk, dinning table, foosball, glass table, kitchen table, office desk, picnic table, poker table, round table, side table, stand, table, vanity, workbench, writing desk

**toilet**: bidet, toilet bowl, toilet seat

**towel**: bath towel, beach towel, face towel, hand towel, paper towel, towel

**tv_monitor**: bulletin board, computer monitor, computer screen, display, monitor, television, whiteboard

### D.4 Mapping Region Class Labels to Tags

We map region labels directly to corresponding tags in the vocabulary of the tagging model whenever possible. Some region labels are made up of multiple concepts such as "porch/terrace/deck/driveway". In these cases, we try to map each concept to a tag if possible.

**balcony**: balcony

**bar**: bar

**bathroom**: bathroom

**bedroom**: bedroom

**classroom**: classroom

**closet**: closet

**dining room**: dining room

**garage**: garage

**hallway**: hallway

**kitchen**: kitchen

**laundryroom/mudroom**: laundry room

**library**: library

**living room**: living room

**meetingroom/conferenceroom**: meeting room

**office**: home office, office

**porch/terrace/deck/driveway**: deck, driveway, porch, terrace

**rec/game**: recreation room

**spa/sauna**: sauna

**stairs**: stairs, stairwell

**tv**: cinema, home theater, theater

**utilityroom/toolroom**: utility room

**workout/gym/exercise**: gym

We relabeled regions labeled "familyroom" and "lounge" as "living room" since these labels described analogous regions. Similarly, we also relabeled regions labeled "toilet" as 'bathroom".

# E   Label to Text Mappings for Embedding-Based Methods

To evaluate embedding-based methods, the class labels are mapped to strings which are then encoded into text embeddings for querying the map embeddings.

## E.1   OpenScene and OpenMask3D

We followed the authors of OpenScene and OpenMask3D and applied the prompting method of "a {} in a scene" for object class labels. We modified the prompt to be grammatically correct for the following object classes:

**chest_of_drawers**: a chest of drawers in a scene

**clothes**: clothes in a scene

**gym_equipment**: gym equipment in a scene

**tv_monitor**: a television monitor in a scene

For region classes, we directly used the label as the string, except for the following classes where we modified the string to either be more grammatically correct or more accurately reflect the class semantics:

**laundryroom/mudroom**: laundry room or mudroom

**meetingroom/conferenceroom**: meeting room or conference room

**porch/terrace/deck/driveway**: porch or terrace or deck or driveway

**rec/game**: recreation or game room

**spa/sauna**: spa or sauna

**tv**: cinema or home theater or theater

**utilityroom/toolroom**: utility room or tool room

**workout/gym/exercise**: gym

## E.2   CLIP Viewpoint Retrieval

For viewpoint retrieval using CLIP, we mapped each label to a string and used the ImageNet80 ensemble prompting method from  Radford et al. [6] to create the text embedding.

For object classes, we used the label directly as the string, except for the following classes where we modified the string to be more grammatically correct or more accurately reflect the class semantics:

**chest_of_drawers**: chest of drawers

**counter**: countertop

**gym_equipment**: gym equipment

**tv_monitor**: television monitor

Similarly for region classes, we used the label directly as the string except for the following classes:

**laundryroom/mudroom**: laundry room or mudroom

**meetingroom/conferenceroom**: meeting room or conference room

**porch/terrace/deck/driveway**: porch or terrace or deck or driveway

**rec/game**: recreation or game room

**spa/sauna**: spa or sauna

**tv**: cinema or home theater or theater

**utilityroom/toolroom**: utility room or tool room

**workout/gym/exercise**: gym

# F  Additional Grounded Navigation Results

The performance of the LLM and Tag Map grounded navigation pipeline is evaluated on a set of user queries that could be asked of a home assistant robot. The evaluation is done on a scan of the lab environment used for the real robot experiments in Sec.5.4. We assume that the environment is entirely traversable such that the robot can reach any goal location from any starting location.

A total of 25 user queries were evaluated. The grounded navigation for a query is considered successful if the LLM grounded on the Tag Map context suggests an instance of a semantic class that is relevant to addressing the query and if that instance is correctly localized from the Tag Map. The Tag Map grounded navigation was successful in 21 of the 25 test queries. The detailed results for all test queries are reported in Tables 9, 10, and 11.

| Query | Suggested Entity | Successful? | Example Goal Viewpoint |
|---|:---:|:---:|---|
| Please heat up my lunch | microwave | ✓ |  |
| Please refill my water bottle | faucet | ✓ |  |
| Please turn on the lights | light switch | ✓ |  |
| Please put away these dirty dishes | dish washer | ✓ |  |
| Can you bring out the ice cream It's the other fridge, not the main one | fridge | ✓ |  |

The LLM localizes two fridges and a kitchen region from the tag map. It selects the fridge further from the kitchen as it associates the fridge in the kitchen as the main fridge.

| Query | Suggested Entity | Successful? | Example Goal Viewpoint |
|---|:---:|:---:|---|
| Ouch, I got a paper cut | soap | ✓ |  |
| Can you go check if there's any mail for me? | mailbox | ✗ |  |

The image tagging model mistakenly recognizes a fire alarm as a mailbox.

| Query | Suggested Entity | Successful? | Example Goal Viewpoint |
|---|:---:|:---:|---|
| Get me something hot to drink | coffee machine | ✓ |  |

Table 9: Tag Map grounded navigation results for the test user queries (1/3).

| Query | Suggested Entity | Successful? | Example Goal Viewpoint |
|---|---|---|---|
| I spilled some water on the floor | `paper towel` | ✓ | |
| I spilled some sand on the floor | `vacuum` | ✓ | |
| Prepare somewhere for me to take a nap | `couch` | ✓ | |
| Can you go downstairs to meet the pizza delivery driver? | `stairwell` | ✓ | |
| It's raining outside, can you bring me something | `umbrella` | ✗ | |

The image tagging model mistakenly recognizes an umbrella in the image.

| | | | |
|---|---|---|---|
| It's getting quite cold in here | `boiler` | ✗ | |

The `boiler` tag describes a kettle, but the LLM misinterprets it as a home heating boiler.

| | | | |
|---|---|---|---|
| It's getting quite cold in here, could you bring me something? | `coat` | ✓ | |

In contrast to the previous query, the LLM interprets this query as bringing an item to warm the user instead of trying to warm the space.

| | | | |
|---|---|---|---|
| Take my phone and charge it | `charger` | ✓ | |
| Help! my shirt is on fire | `extinguisher` | ✓ | |
| Please prepare a location for the party this evening | `terrace` | ✓ | |

Table 10: Tag Map grounded navigation results for the test user queries (2/3).

| Query | Suggested Entity | Successful? | Example Goal Viewpoint |
|---|---|---|---|
| The sunlight is making it too bright in here | blind | ✓ |  |
| Get me something to read | magazine | ✓ |  |
| Can you get things ready for the barbeque later | barbeque grill | ✗ |  |

The image tagging model mistakenly recognizes a large metallic object as a barbeque grill.

| Query | Suggested Entity | Successful? | Example Goal Viewpoint |
|---|---|---|---|
| Set up a game for me and my friend to play | table tennis table | ✓ |  |
| Please take out the trash | bin | ✓ |  |
| Can you get some fresh air in here? | window | ✓ |  |
| Can you bring me something healthy to eat | fruit | ✓ |  |

Table 11: Tag Map grounded navigation results for the test user queries (3/3).

# G  Qualitative Comparison of Tag Map and Embedding Maps Localizations

We present a selection of qualitative comparisons between the localization produced by the Tag Map, OpenScene, and OpenMask3D. The selected examples do not represent an exhaustive selection but rather aim to provide examples of localization successes and failures for the methods.

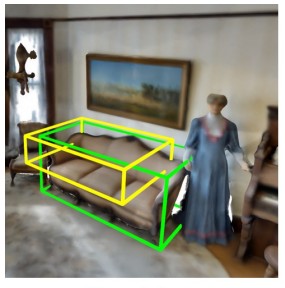 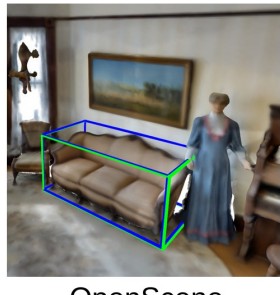 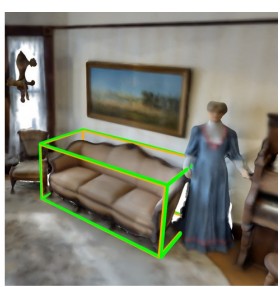

| Tag Map | OpenScene | OpenMask3D |

Figure 9: Localizations for `sofa`. The Tag Map can only localize the sofa coarsely, while the embedding-based maps store more geometric information and can localize it more precisely.

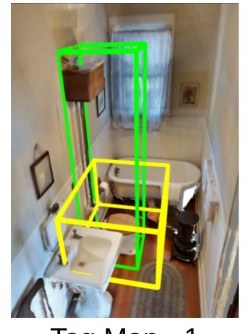 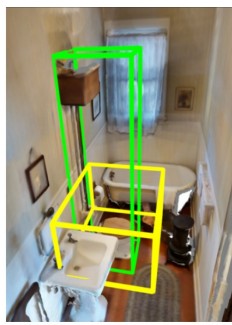 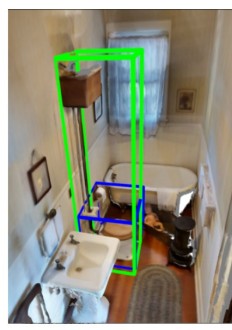 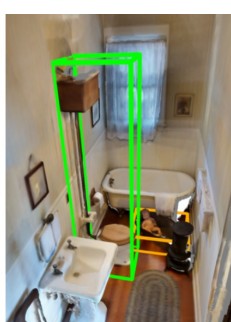

| Tag Map - 1 | Tag Map - 2 | OpenScene | OpenMask3D |

Figure 10: Example localizations for `toilet`. In this case, we show two localizations produced by the Tag Map. The second localization has worse P2E compared to the first since a significant portion of it is outside the room of the toilet.

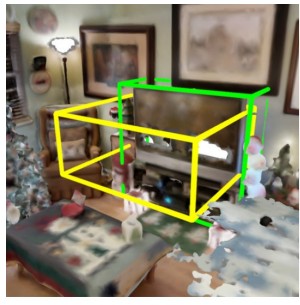 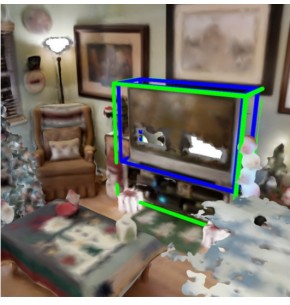 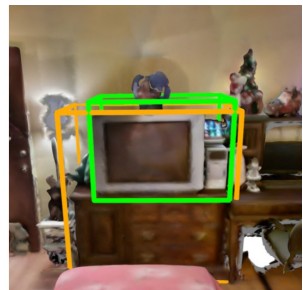

| Tag Map | OpenScene | OpenMask3D |

Figure 11: Examples of correct localizations for `tv_monitor`.

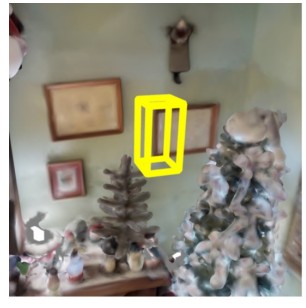 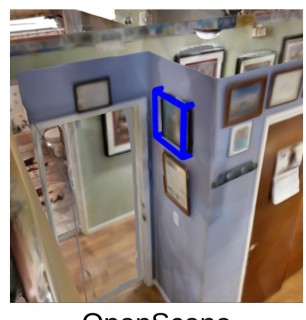 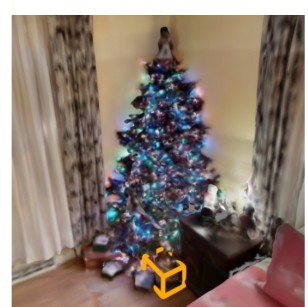

Tag Map        OpenScene        OpenMask3D

Figure 12: Examples of wrong localizations for `tv_monitor`. The Tag Map produces a wrong localization as the pictures on the wall are falsely recognized as a screen by the tagging model. Similarly, OpenScene labels a picture on the wall as a screen. OpenMask3D wrongly labels an instance mask.

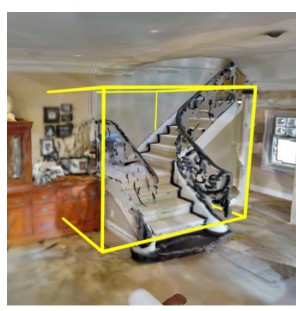 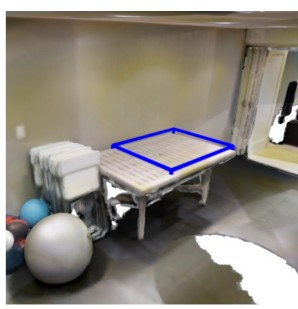 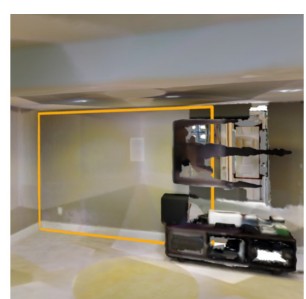

Tag Map        OpenScene        OpenMask3D

Figure 13: Examples of wrong localizations for `bed`. The Tag Map produces a wrong localization due to a false positive detection of the tag `bed`. OpenScene wrongly labels a table that looks like a bed. OpenMask3D wrongly labels a wall as a bed.

