# OpenReview forum: "Tag Map: A Text-Based Map for Spatial Reasoning and Navigation with Large Language Models"
_robot-learning.org/CoRL/2024/Conference — CoRL 2024_

### Official Review · Reviewer_99N9 · 2024-07-16
**A nice novel and compact representation of the scene**

**Originality:** 5
**Technical Quality:** 2
**Clarity Of Presentation:** 5
**Potential Impact:** 3
**Recommendation:** 3
**Confidence:** 5

**Review:**

Strengths:
1. In short, the idea proposed in the paper is novel and the experiment showed its effectiveness and compactness compared to open-vocabulary mapping alternatives like OpenScenes and OpenMask3D. However, I have a few doubts regarding the evaluation which will be described in the next session.
2. The flow of the paper is excellent and the idea is concisely described. However, some design details are missing from the paper or appendix.

Weakness and questions:
1. The idea of using overlapped frustum to represent objects is indeed compact but it has two major problems: (1) the frustum is prone to be under-segmented in 3D, meaning that the points inside the overlapped frustum tend to contain irrelevant artifacts, and (2) the quality of the overlapped frustum is highly dependent on the directions and positions of those multi-views associated with the same tag. Regarding (2), for example, you need to view objects from more than 2 views with large baselines to have constraints on the 3D space for those 2D tags. If an object is viewed only once which might happen during exploration, the frustum might represent a large space that is associated with a certain tag. As a result, the method might work well in a closed space where the objects are close to the wall which helps constrain the frustum but might fail easily in open and clutter space with objects in the center. This implicit viewpoint assumption is mentioned in the limitations but I think it is a major issue of the method and fatal to the generalization ability of the method.
2. The metrics P2E and E2P are not fair and biased. Undersegmentation or undersegmentation are preferred in this metric. Imagine that the predicted point cloud (or bbox) is larger than the desired target point cloud (or bbox) and is the superset of the target, the path connecting the two will be 0. So is the other way around. I would like to see the 3D bounding box precision or recall, and the navigation (or planning) success rate from a random position to the retrieved object (stop position <1 meter) in the simulator which are reproducible.
3. Tables 1 and 2 do not mention P2E and E2P which are introduced in Sec. 4.1. I don't know what distance is used in the table. I hope the author can address this confusion by adding captions under the table or mentioning it in the experiment part when introducing the table.
4. I expect some qualitative results comparing OpenScenes, OpenMask3D, and the proposed method like visualizing the bounding boxes of some instances. I think it might be also a good way to help authors figure out the potential drawbacks of the method.
5. I wonder why the authors only limited the semantic classes to 21 classes in Matterport3D which supports 40 categories. Excluding some meaningless labels, there are still at least 30 meaningful categories in the dataset. I hope the authors can justify their choice.
6. More details regarding the depth statistics are needed in the paper or appendix. Line 106-107 say "mapping each view106 point to its 3D frustum using the stored depth statistics." but I think it is a bit ambiguous. What is depth statistics and how is it exactly done?

**Quality Of The Limitations Section:**

2

**Questions For Rebuttal:**

The short questions listed here are corresponding to the bullet points in the review weakness section. Please refer to the review section for more details and answer them:
1. The method is prune to under-segmentation and the quality of the overlapped frustum is highly dependent on the viewing points. How did you solve this?
2. The metric of P2E and E2P are not fair and biased. Correct me if I am wrong. But I would also like to see the 3D bounding box precision or recall, and the navigation (or planning) success rate from a random position to the retrieved object (stop position <1 meter) in the simulator which are reproducible.
3. Tables 1 and 2 do not mention P2E and E2P. What is the threshold distance used in the tables? State it in the caption.
4. Is it possible to show some qualitative results comparing to OpenScenes and OpenMask3D?
5. Why do the authors only consider 21 categories among 40 in Materport3D?
6. Explain more regarding line 106-107.

**Robotics Focus:**

4

**Summary Of Paper:**

The authors proposed to associate text tags and their corresponding bounding boxes generated in 2D with overlapping 3D view frustums to create a semantic scene representation.

**Summary Of Recommendation:**

If the authors can address my concerns mentioned in the review, I will raise the recommendation to one level higher.

---

### Official Review · Reviewer_Bsyi · 2024-07-17
**Review of Tag Map: A Text-Based Map for Spatial Reasoning and Navigation with Large Language Models**

**Originality:** 3
**Technical Quality:** 3
**Clarity Of Presentation:** 4
**Potential Impact:** 3
**Recommendation:** 3
**Confidence:** 3

**Review:**

# Quality
The paper is well-polished. The problem of using implicit tag maps rather than explicit tag maps is defined in the beginning, and the paper establishes the context of the problem in the introduction. The method is accompanied by benchmarks and real-world experiments with a quadrupedal robot navigating a building floor. Comparisons are made against existing embedding-based tag maps. Ablation tests are also performed on the components of the tag map generation pipeline.

# Clarity
The paper explains the motivation for the project, as well as the methods used in the approach. The process of creating the tag map is explained, including the step involving localizing the tags with respect to the map viewpoints. The paper defines the metrics used in evaluating coarse-grained localization. Limitations could use some more elaboration, such as the presence of false-positive tags.

# Originality
The paper makes use of existing models in order to generate the data structure, and it also uses an existing LLM to evaluate the effectiveness of the model. The novelty comes from the data pipeline that takes RGB-D data as input and produces the tag map. GPT-4 is used as the LLM that calls the exposed API of the tag map.

# Significance
An explicit text map allows the use of separately-trained models to use the data structure without additional training on implicit embeddings. The approach can expose an API for LLMs to query in order to obtain positional data for text queries.

# Strengths
- Project is readily applicable to other uses in robotics and AI.
- Straightforward method makes implementation trivial.

# Weaknesses
- The paper mainly reuses existing models to generate the text map and does not feature any new model architectures.

**Quality Of The Limitations Section:**

2

**Questions For Rebuttal:**

How often does the approach produce false positive tags? Are the false positives more common with certain tags than others?

**Robotics Focus:**

4

**Summary Of Paper:**

The paper presents a method to generate a data structure of localized tags from scene data. Semantic classes are obtained from the map through localization. The generated map is then used as context for a LLM in order to assist in navigation of a robot.

**Summary Of Recommendation:**

The paper applies existing models to produce a useful data pipeline to produce textual maps of scenes. This has obvious applications in robot planning and as queries for LLMs. No new models were presented in the paper.

---

### Official Review · Reviewer_W8CK · 2024-07-23
**Well written paper introducing a novel approach for text-based semantic mapping and localization in unseen environments for object goal navigation tasks. Answers to some questions regarding the inference time of this approach, utility in real-time applications, incorporating feedback and comparisons to 3D scene graph based methods are desired.**

**Originality:** 4
**Technical Quality:** 3
**Clarity Of Presentation:** 4
**Potential Impact:** 3
**Recommendation:** 3
**Confidence:** 3

**Review:**

Strengths:

The paper is well written and provides an interesting new approach for building text-based maps with thousands of  semantic classes.

Extensive evaluations and comparison to competitive baselines show that these text-based maps perform comparably to open vocabulary maps while using orders of magnitude less memory

Real world experiments in  a lab/office scene using the legged robot ANYmal demonstrate how a user can interact with the LLM and answer clarifying questions

The authors clearly discuss some limitations of this approach and interesting directions for future work

Ablations are very useful in understanding the different components of the method


Weaknesses:

It seems like, in the proposed approach, using LLMs for planning is not necessarily needed. If the task is clearly specified with both the region and the entity, the relevant tags can be directly extracted from the tag-map. The authors point out that they focus on tasks “where the relevant entities for solving the task are not explicitly mentioned and must be inferred by the LLM through the tag map context”. This is a fair assumption but not critical to the overall method.

Although it is advantageous to have low memory usage, a much more desirable feature for real-time grounded navigation tasks can be low inference time. A discussion on the how much slower is this approach compared to prior works (OpenScene, OpenMask3D) could be useful

**Quality Of The Limitations Section:**

2

**Questions For Rebuttal:**

Can the proposed text-based map be generated in real time? Additionally, given a partially constructed map can a good course localization be done if the object is detected and terminate exploration? Furthermore, can an LLM be used to direct exploration to relevant parts of the scene using the partially built tag-map?

How long does it take to generate the course localizations and plans using the proposed method as compared to prior work. Can it be used for real-time navigation tasks?

If the proposed localization and plan turns out to be incorrect what are effective/efficient ways of replanning?

Can the proposed approach be extended to control in continuous environments (VLN-CE tasks) i.e. without relying on matterport3D viewpoints? And, how?

How does this method compare to 3D scene graph representations such as ConceptGraphs, which are also open-vocabulary, object-centric and compact?

**Robotics Focus:**

4

**Summary Of Paper:**

The paper proposes a method for building text-based maps of unknown scenes for object goal navigation tasks. The map consists of viewpoint IDs and associated semantic classes recognized in the image by a multi-label classification model. Course localization of a desired tag is done by reconstructing the 3D geometry using RGBD images at the tag’s corresponding viewpoints. Finally, an LLM, equipped with localization and relative distance calculation APIs, can be prompted with the tag map to find a path to the desired tag. Course localization from these text-based maps perform comparably to open vocabulary maps while using orders of magnitude less memory.

**Summary Of Recommendation:**

The work presents an interesting new approach for building text-based maps with thousands of semantic classes. Localization is done only for the desired tags thus requiring much less memory. This is a novel approach that can be useful for object-goal navigation tasks.

---

### Official Review · Reviewer_4KgS · 2024-08-01
**Well Written and Well Motivated Paper with Some Missing Evaluations**

**Originality:** 3
**Technical Quality:** 3
**Clarity Of Presentation:** 4
**Potential Impact:** 3
**Recommendation:** 3
**Confidence:** 3

**Review:**

The paper is written quite clearly and is well motivated. Tag maps have significantly lower memory footprint compared to embedding based maps and I really appreciate the thorough nature of the ablations and the analysis of the precision/recall of localization from tagmap vs. a CLIP-based retrieval. The robot results on the ANYmal platform are also nice, but lack sufficient detail for me to understand whether the stated improvements from tagmap actually manifest when performing grounded navigation.

It is however a bit strange that tag maps exhibit such low performance in terms of recall on region classes but such good recall for objects. Further discussion regarding this discrepancy would be much appreciated. A big weakness of the paper is that there are no quantitative results provided regarding how often the tagmap based navigation system (when integrated with the LLM) is actually able to reach the desired regions. There is a nice qualitative example showing how tag map is applied to navigation on the ANYmal platform, but there are no quantitative results evaluating the full system. I feel that this is quite important for this work, because tag map does not uniformly outperform embedding based maps in terms of precision/recall for both objects and region classes. As a result, it is critical to evaluate whether tag map can lead to concretely better (or at least similar) navigation results when combined with an LLM planner compared to existing embedding based maps.

**Quality Of The Limitations Section:**

3

**Questions For Rebuttal:**

1. Could you expand on why exhibit such low performance in terms of recall on region classes but such good recall for objects?
2. Can you provide concrete quantitative results evaluating how often tagmap based planning leads to the robot navigating to the correct location in either simulation on real experiments?
3. Can you repeat #2 with embedding-based mapping techniques to establish that tagmap can at least maintain their performance while achieving a lower memory footprint?

If the above points are addressed, I would be very happy to update my score.

**Robotics Focus:**

4

**Summary Of Paper:**

This paper proposes a memory efficient text based map for storing recognized entities and their viewpoints as text tags. This information is sufficient to produce 3D localizations for objects, rooms, and regions at sufficient granularity for vision-based navigation. These localizations, when combined with an LLM provided with a simple API interface, enables flexible generation of navigation plans given a task specification from a user.

**Summary Of Recommendation:**

Generally, I am positive on the paper and think that tagmap is well motivated and explained quite clearly. The main issue with the paper is the lack of quantitative metrics indicating whether tagmap can achieve similar grounded navigation results when integrated with an LLM-based planner when compared to embedding based maps. If this comparison can be clearly established quantitatively, then I am happy to update my score, but as it stands I am on the fence between Accept/Reject, but leaning Accept.

---

### Author Rebuttal · Authors · 2024-08-08

The rebuttal materials contain the revised paper which includes the following changes:
- added more details on the depth statistics based on the feedback from Reviewer 99N9
- added more explanation for the terms "Precision at Threshold" and "Recall at Threshold" used in the tables based on feedback from Reviewer 99N9

It also contains a revised appendix which includes the following changes:
- added quantitative evaluations of the tag map grounded navigation pipeline on a set of user queries based on the feedback from Reviewer 4KgS
- added qualitative results for comparing the localizations of the tag map, openscene, and openmask3d based on feedback from Reviewer 99N9

---

### Decision · Program_Chairs · 2024-09-04

**Decision:**

Accept

**Comment:**

Update:
After the rebuttal, the reviewers are in consensus about the paper's novelty, contribution, and execution. The proposed method works well and seems easy to adopt by others. I will recommend acceptance for this paper.

Original:
The reviewers find the proposed method novel and interesting, but they share some concerns regarding its generalizability under different scenarios as well as various experimental details. I ask the authors to carefully address all reviewer comments during the rebuttal period.